# Local amphotericin B therapy for Cutaneous Leishmaniasis: A systematic review

**Líndicy Leidicy Alves** *, **Mariana Lourenço Freire, Isadora Lana Troian, Eliane de Morais-Teixeira, Gláucia Cota**

Clinical Research and Public Policy Group on Infectious and Parasitic Diseases–René Rachou Institute—Fundação Oswaldo Cruz–FIOCRUZ, Belo Horizonte, Minas Gerais, Brazil

* lindicy.leidicy@hotmail.com

**Data Availability Statement:** All relevant data are within the manuscript and its Supporting Information files.

## Abstract

### Background

Cutaneous leishmaniasis (CL) is characterized by potentially disfiguring skin ulcers carrying significant social stigma. To mitigate systemic drug exposure and reduce the toxicity from available treatments, studies addressing new local therapeutic strategies using available medications are coming up. This review systematically compiles preclinical and clinical data on the efficacy of amphotericin B (AmB) administered locally for cutaneous leishmaniasis.

### Methodology

Structured searches were conducted in major databases. Clinical studies reporting cure rates and preclinical studies presenting any efficacy outcome were included. Exclusion criteria comprised nonoriginal studies, *in vitro* investigations, studies with fewer than 10 treated patients, and those evaluating AmB in combination with other antileishmanial drug components.

### Principal findings

A total of 21 studies were identified, encompassing 16 preclinical and five clinical studies. Preclinical assessments generally involved the topical use of commercial AmB formulations, often in conjunction with carriers or controlled release systems. However, the variation in the treatment schedules hindered direct comparisons. In clinical studies, topical AmB achieved a pooled cure rate of 45.6% [CI: 27.5–64.8%; I2 = 79.7; p = 0.002], while intralesional (IL) administration resulted in a 69.8% cure rate [CI: 52.3–82.9%; I2 = 63.9; p = 0.06]. In the direct comparison available, no significant difference was noted between AmB-IL and meglumine antimoniate-IL administration (OR:1.7; CI:0.34–9.15, I2 = 79.1; p = 0.00), however a very low certainty of evidence was verified.

### Conclusions

Different AmB formulations and administration routes have been explored in preclinical and clinical studies. Developing therapeutic technologies is evident. Current findings might be interpreted as a favorable proof of concept for the local AmB administration which makes

**Funding:** LLA 001 by the Coordenação de Aperfeiçoamento de Pessoal de Nível Superior (CAPES), available at https://www.capes.gov.br. MLF 151891/2022-2 by the Conselho Nacional de Desenvolvimento Científico e Tecnológico (CNPq), available at http://www.cnpq.br. GC 302069/2022-4 by the Conselho Nacional de Desenvolvimento Científico e Tecnológico (CNPq), available at http://www.cnpq.br. The funders had no role in study design, data collection and analysis, decision to publish, or preparation of the manuscript.

**Competing interests:** The authors have declared that no competing interests exist.

this intervention eligible to be explored in future well-designed studies towards less toxic treatments for leishmaniasis.

## Author summary

Cutaneous leishmaniasis (CL) is a neglected disease recognized for causing ulcers distributed throughout the body; such ulcers can be disfiguring and may intrinsically attract a social stigma. Furthermore, the currently available therapeutic arsenal is admittedly limited and associated with significant toxicity. Long therapeutic regimens and parenteral administration have been considered in recent years as a strategy to be replaced. In this scenario, the search for new treatment alternatives for CL is considered a priority by the World Health Organization, which has been expanding its treatment recommendations with alternatives for local use. In this study, we performed an extensive literature search to gather evidence on the efficacy of amphotericin B (AmB) in local administration for the treatment of CL in preclinical or clinical studies. In total, 21 studies, 16 preclinical studies and five clinical studies were identified. The global cure rate for clinical studies addressing intralesional infiltration and topical application was estimated at 56.9% (CI: 41.1–71.4%). High heterogeneity in the design, groups, and route of administration of AmB among studies was observed, requiring caution in interpreting and extrapolating of this pooled rate. In summary, the results show that local routes of administration of AmB are a promising strategy and need to be evaluated in studies with adequate design, enabling the expansion of therapeutic alternatives for the treatment of CL.

## Introduction

Leishmaniasis is an infectious parasitic disease that occurs in tropical low-income countries, usually with limited access to health care. The clinical manifestation varies according to the species of *Leishmania*, immune response and, probably, other concomitant host conditions. Compared to the visceral and mucosal or mucocutaneous leishmaniasis forms (VL and MCL), cutaneous leishmaniasis (CL) is the most prevalent, with more than 220 thousand new cases worldwide [1,2].

Characterized by non-fatal but potentially disfiguring ulcers distributed across the body, CL is linked with social stigma persisting post-treatment due to the risk of permanent scarring and relapses [3,4]. The optimal treatment is not yet available for leishmaniasis, and meglumine antimoniate and amphotericin B (AmB), both of which are related to significant toxicity, are the most used drugs [5].

AmB is recognized for its high leishmanicidal effect. Initially, made available in a formulation using sodium deoxycholate as a solubilizing agent, this drug has also become known for its potential for nephrotoxicity [6]. With a focus on reducing the occurrence of this significant adverse event, lipid formulations of AmB were developed, including liposomal AmB (L-Amb), a presentation based on AmB incorporation into liposomes, which still traditionally requires intravenous administration [7,8]. A further approach to mitigate the undesirable effects of medication involves modifying the route of administration. This rational has been employed in the management of CL, wherein pentavalent antimony was previously administered parenterally and now receives a strong endorsement for intralesional application [9]. The benefits of intralesional infiltration primarily stem from the utilization of lower amount of antimony,

resulting in a notable reduction in its associated toxicity [10]. Additionally, the topical or intralesional administration of a drug is anticipated to streamline the treatment process, concurrently diminishing the complexity of intravenous administration and the costs related hospitalization [11]. This review systematically aggregates data on the treatment of cutaneous leishmaniasis with locally administered AmB, encompassing both clinical and preclinical studies. The primary aim is to map the current level of evidence, providing the groundwork for a new therapeutic approach development plan and design of future clinical trials [12].

## Methods

### Protocol and registration

The protocol of this review was registered in the International Prospective Register of Systematic Reviews (PROSPERO: CRD42021265854). This systematic review was conducted according to the Preferred Reporting Items for Systematic Reviews and Meta-Analyses (PRISMA—S1 Table) [13].

### Eligibility criteria

The systematic review was guided by the following research question: 'What is the efficacy of local treatment with amphotericin B for cutaneous leishmaniasis?' Adhering to the PICOS framework (population, intervention, comparator, outcome, study design), the review selection process followed the specific inclusion criteria: (P) patients with cutaneous leishmaniasis or animals infected with species of Leishmania causing cutaneous leishmaniasis, (I) utilization of local treatment with AmB, (C) other therapeutic interventions, placebo, or no control group, (O) cure rates in clinical studies or any other efficacy outcome for animal studies, and (S) original studies without design restrictions. This rigorous approach aimed to systematically assess and synthesize relevant literature, ensuring a comprehensive exploration of the efficacy of AmB in the local treatment of cutaneous leishmaniasis. Exclusion criteria were as follows: (I) nonoriginal studies, including literature reviews, editorials, brief communications, and case reports; (II) in vitro studies; (III) studies with fewer than 10 treated patients; and (IV) studies evaluating AmB combined with another antileishmanial active component. There were no language or publication date restrictions.

### Search strategy

Structured searches were performed in MEDLINE (accessed by PubMed), Latin American and Caribbean Literature on Health Sciences (accessed by Virtual Health Library), Excerpta Medica Database (Embase), Cochrane Library and Web of Science. Search strategies combining the keywords related to cutaneous leishmaniasis AND AmB were constructed for each database, and the strategies are summarized in the S2 Table. A manual search was also performed by analysis of the reference lists of selected articles. All searches were performed up to October 30, 2022.

### Selection process, data extraction and outcomes

For each database, all recovered articles were added to the Mendeley reference manager for duplicate citation exclusion. According to the inclusion and exclusion criteria, two independent reviewers (LLA and MLF) analyzed each publication by title and abstract using Rayyan software [14]. Disagreements were resolved by consensus or with a third reviewer (GC). Data were extracted from the included studies by two independent reviewers (LLA and MLF).

For preclinical studies, methodological characteristics of the included studies, such as animal models, *Leishmania* species, infection sites, treatment schedules, comparator groups, outcomes of interest and follow-up were extracted.

For clinical studies, standard data extraction forms were used to collect the main study and population characteristics and outcomes. For these studies, the outcome of interest was the initial cure rate. Considering D1 as the first day of treatment and using the definitions proposed by Olliaro (2013) [15], which establishes that initial cure should be evaluated around D90, here we assumed as the initial cure rate that assessed between D42 and D90, as an adequate range capable of encompassing the greatest number of studies. The cure rate was expressed as the number of cases cured by the total number of cases treated with a given intervention (intention-to-treat approach). Losses observed during follow-up were considered therapeutic failures. To estimate the relapse rate, only patients considered previously cured were included. The safety of antileishmanial therapy was captured in each study as the number of adverse events per total number of evaluated patients (or treatments, if this was the only information available).

## Data synthesis and statistical analysis

Comprehensive Meta-Analysis software v.3.0 was used to perform a one-group meta-analysis of study arms using a given treatment (pooled rates) based on an assessment of the baseline cure rate, as arbitrarily adopted (D90), reported in the original studies. Clinical cure rates were calculated according to the intention-to-treat approach, that means that the analysis was based on the total number of randomly assigned participants, irrespective of how the original study's authors analyzed the data. We used the inconsistency ($I^2$) statistic to evaluate heterogeneity, using the following interpretation for I2: up to 40%: low; 30 to 60%: moderate; 50 to 90%: substantial; 75 to 100%: considerable heterogeneity [16].

## Study quality assessment

The assessment of evidence quality was done using the recommended tools tailored to the specific study designs. For animal studies, the SYRCLE risk, an adapted iteration of the Cochrane RoB tool of bias, was utilized [17] Nonrandomized clinical trials had the evidence quality assessed throught the Newcastle Ottawa Scale (NOS) [18], while randomized controlled trials were subjected to the Cochrane risk of bias score (RoB 2) [16]. All evaluation were performed by two independent researchers (LAA and EMT for preclinical studies and LAA and MLF for clinical studies). Discrepancies were resolved through consensus, or by a third reviewer (GC) in the absence of consensus.

## Assessment of certainty in evidence

The certainty of evidence was evaluated through the GRADE tool [19], which guides the assessment of risk of bias across multiple domains, namely indirect evidence, inconsistency, imprecision, and publication bias. Through the application of this tool, the quality of evidence can be classified into four levels: high, moderate, low, and very low.

## Results

### Literature search

A total of 8,583 studies were recovered from MEDLINE (1,952), LILACS (253), Cochrane Library (25), Web of Science (2,261) and Embase (4,092). Initially, all duplicates were removed, and 4,911 titles and abstracts were analyzed. Of the 73 studies selected for reading in

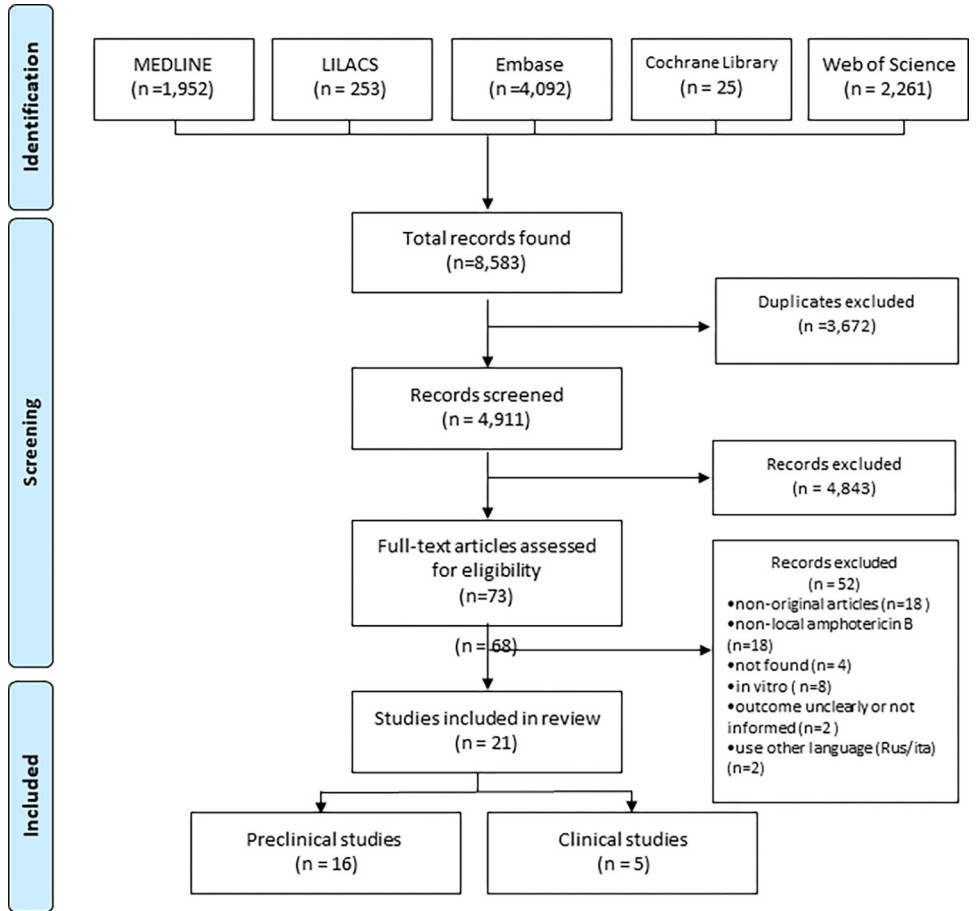

**Fig 1. PRISMA flow diagram of the study selection process.**

full, 21 were included: 16 preclinical and five clinical studies. The 52 excluded studies during the full reading are listed in the S3 Table. The study selection process is summarized in the PRISMA flow diagram (Fig 1).

## Preclinical studies

Sixteen included studies using animal models are presented in Table 1. Four studies were carried out in Brazil [20–23] and the United Kingdom [23–26]. Other studies were conducted in the USA [27,28], Iran [27,29,30] and Israel [31–33].

The *Leishmania* species evaluated in studies were *L. (L.) major* (12), *L. (L.) mexicana* (3) and *L. (L.) amazonensis* (2). Nguyen et al., 2019 [28] evaluated both *L. (L.) major* and *L. (L.) mexicana*. As expected, the BALB/c mouse (*Mus musculus*) was the most reported experimental model, and the infection was performed at the base of the animal's tail (13 studies).

The cure definitions adopted by the authors also varied significantly. For most studies, efficacy was presented not as the cure rate (the proportion of cured animals) but as the magnitude of the lesion size reduction and/or parasitological clearance (in the lesion tissue and/or spleen). In general, wide variation was observed among the treatment protocols and outcomes reported, making it difficult to compare the results in these different studies

**Table 1. Main characteristics of the animal studies.**

| Year, Author | Infection models | | | Treatment | | | Outcome | | |
|---|---|---|---|---|---|---|---|---|---|
| | Animal | Infection site/ *Leishmania* species of infection | Animals/ group | Start of treatment | Intervention/ administration route treatment schedule | Comparator | Methodology | Follow-up | Intervention X Comparator |
| El-On, 1984 [31] | BALB/ C | Tail base/ *L. (L.) major* | 5 | 35 days post-infection | Fungisome with methyl benzethonium chloride/topical twice daily for 12 days | The lesion size before the treatment | lesion size | 30 days after end of treatment | There was no reduction in the lesion size or cure of any animals. |
| Yardley, 1997 [24] | BALB/ C | Tail base/ *L. (L.) major* | NR | 21 to 28 days post-infection | AmBisome/ subcutaneous 25 mg/ kg once a day on six alternate days | Methylcellulose (negative control) | lesion size | 35 days after end of treatment | Nonsignificant reduction. |
| Frankenburg, 1998 [32] | Mouse CBA | Tail base *L. (L.) major* | 8 | 24 h post-infection | 10 μl of Amphocil (2 mg/ml) topical, daily for 3 weeks; (I) in glycerol (II) in polypropylene glycol (III) in 10% ethanol (IV) in glucose | Untreated | lesion size | 35 days post-infection | Statistically significant reduction in the lesion size only for Amphocil with 10% ethanol. |
| | | | | 24 h post-infection | 10 μl of Amphocil (2 mg/ml) topical, daily for 3 weeks; (I) in 5% ethanol; (II) in 10% ethanol; (III) in 25% ethanol | | lesion size | 28 days post-infection | Reduction in lesion size regardless of the ethanol. |
| | | | | 24 h post-infection | (I) 10 μl of Amphocil (2 mg/ml) in 5% ethanol (topical), daily for 3 weeks; (II) 10 μl of Fungizone (2 mg/ml) in 5% ethanol (topical), daily for 3 weeks; (III) 10 μl of Amphocil (2 mg/ml) in double-distilled water (topical), daily for 3 weeks; | | lesion size | 28 days post-infection | The lesion size reduction with Fungizone is less than that with Amphocil. |
| Corware, 2010 [25] | BALB/ C | Footpad/ *L. (L.) major* | 24 | 7 days post-infection | AmB-polymethacrylic acid)/subcutaneous 2 mg/kg each day (7, 14 and 21 post infection day) | Water subcutaneous | lesion size/lesion's parasitic count with a hemocytometer | 80 days post-infection | 35 days post-infection: 82 ± 2% lesion reduction and 99.85 ± 0.03% reduction in parasite viability/50 days post-infection: complete lesion healing/80 days post-infection: no relapse. |

**Table 1.** (Continued)

| Year, Author | Infection models | | | Treatment | | | Outcome | | |
|---|---|---|---|---|---|---|---|---|---|
| | Animal | Infection site/ *Leishmania* species of infection | Animals/ group | Start of treatment | Intervention/ administration route treatment schedule | Comparator | Methodology | Follow-up | Intervention X Comparator |
| Corware, 2011 [26] | BALB/ C mice | Footpad/ *L. (L.) major* | 18 | 7 days post-infection | AmB-polymethacrylic acid/ intradermal 6 mg/kg each day (7, 14 and 21 post infection day) | Water | lesion size/lesion's parasitic count with a hemocytometer | 35 days post-infection | Resolution of cutaneous lesions and decrease in the parasite number by >3 log10 and 35 to 80 days post-infection: no relapse. |
| | | | 16 | 21 days post-infection | AmB-polymethacrylic acid/ intradermal 6 mg/kg each day (21, 25 and 28 post infection day) | | | 80 days post-infection | |
| Pinheiro, 2016 [20] | BALB/ C | Tail base/ *L. (L.) major* | 8 | at least nodules in most animals and before ulcerative lesions | AmB 3% + Emulgel/ topical 50 µL twice daily for 12 days | Emulgel topical | lesion size/lesion's parasitic load: no description of the methodology | 35 days after the end of the treatment | Ulcerative lesions regressed gradually to even a complete cure/larger reduction in the number of recovered parasites. |
| | | | | | AmB 3% + Emulgel + Oleic Acid 5%/ topical 50 µL twice daily for 12 days | | | | Ulcerative lesions regressed gradually to even a complete cure/larger reduction in the number of recovered parasites. |
| Varikuti, 2017 [27] | BALB/ C | Tail base/ *L. (L.) mexicana* | NR | 56 days post infection | SinaAmphoLeish 0.4% (Nanoliposomal AmB)/topical twice daily for 28 days | Vaseline topical | lesion size/lesion's parasitic load determined by limiting dilution | 15-week post-infection | No significant differences in the lesion sizes and parasitic burdens. |
| | Mouse 129SVE | Tail base/*L. (L.) mexicana* | NR | 35 days post infection | SinaAmphoLeish 0.4% (Nanoliposomal AmB)/topical twice daily for 70 days | Vaseline topical | lesion size/lesion's parasitic load determined by limiting dilution | 15-week post-infection | Transient decrease in lesion sizes during the treatment but no significant differences in the lesion sizes and parasitic burdens. |
| Abu Ammar, 2019 [33] | BALB/ C | Tail base/ *L. (L.) major* | 6 or 7 | 73 days post-infection | Poly(lactic-co-glycolic acid) nanoparticles (NPs) loaded with AmB deoxycholate/ intralesional 1 mg/kg single dose | Control PBS; Blank NPs; AMB deoxycholate 1 mg/ kg intralesional | lesion size | 34 days after star treatment | AMB NPs elicited a significantly greater lesion-reducing effect than the controls. |

(*Continued*)

**Table 1.** (Continued)

| Year, Author | Infection models | | | Treatment | | | Outcome | | |
|---|---|---|---|---|---|---|---|---|---|
| | Animal | Infection site/ *Leishmania* species of infection | Animals/ group | Start of treatment | Intervention/ administration route treatment schedule | Comparator | Methodology | Follow-up | Intervention X Comparator |
| Jaafari, 2019 [29] | BALB/ C | Tail base/ *L. (L.) major* | 10 | 28 days post-infection | Nanoliposomal Amphotericin B 0.1% (50 mg) topical/twice daily for 28 days | PBS; Empty liposomes topical/twice daily for 28 days | lesion size/lesion and spleen parasitic load determined by limiting dilution | 12-week post-infection | Remarkable reduction in the lesion size in treated animals from week 8 onward, with no significant difference among different groups receiving AmB nanoliposomal/The splenic and lesion parasitic load in group treated with Lip-AmB 0.4% was significantly lower than control groups at week 12 post infection. |
| | | | | | Nanoliposomal AmB 0.2% (50 mg) topical/ twice daily for 28 days | | | | |
| | | | | | Nanoliposomal AmB 0.4% (50 mg) topical/ twice daily for 28 days | | | | |
| Nguyen, 2019 [28] | BALB/ C | Tail base/ *L. (L.) mexicana* | NR | 35 days post-infection | AmB DMSO/topical 25 mg/kg/day once a day for 10 days | (I) PBS vehicle (II) AmBisome intraperitoneal 20 mg/kg | lesion size/lesion´s parasitic load determined by limiting dilution | 21 days after treatment | In lesion size significant differences in relation AmB dissolved in DMSO topical (p<0.05)/In the 500 μm needle group was the lowest parasitic load recorded in this experiment. |
| | | | | | AmB DMSO/topical with microneedling 500 μm 25 mg/kg/day once a day for 10 days | | | | |
| | | | | 35 days post infection | AmB DMSO/topical microneedling 750 μm 25 mg/kg/day once a day for 10 days | | | | |
| | | | | | AmB DMSO/topical with microneedling 1000 μm 25 mg/kg/ day once a day for 10 days | | | | |
| | | Tail base/ *L. (L.) major* | | | AmB DMSO/topical 25 mg/kg/day once a day for 10 days | (I) Saline vehicle 0,9% intraperitoneal; (II) microneedles 750 μm individually; (III) AmBisome intraperitoneal; topical AmB DMSO | | 14 days after treatment | There was no difference in the lesion size or parasitic load between the group receiving AmpB after microneedle and any of the non-AmBisome groups. |
| | | | | | AmB DMSO/topical with microneedling 750 μm 25 mg/kg/day once a day for 10 days | | | | |

(*Continued*)

**Table 1.** (Continued)

| Year, Author | Infection models | | | Treatment | | | Outcome | | |
|---|---|---|---|---|---|---|---|---|---|
| | Animal | Infection site/ *Leishmania* species of infection | Animals/ group | Start of treatment | Intervention/ administration route treatment schedule | Comparator | Methodology | Follow-up | Intervention X Comparator |
| Sousa-Batista, 2019 [21] | BAlb/C | Ear pinnae/*L. (L.) amazonesis* | NR | 10 or 30 days post-infection | Fungizone and polylactic-co-glycolic acid (PLGA)/ intralesional/single dose 5 μg (0.2 mg/kg) of AmB in Day 10 (the model of early CL) | PBS | lesion size; parasitic load; Measurement of the lesion size/ lesion and lymph node parasitic load determined by limiting dilution | 120 days after infection | At the end of follow up, lesions were 37% smaller, and the parasite burdens in the ear and draining lymph nodes were 85% and 78% smaller than PBS control, respectively. |
| | | | NR | | Fungizone with polylactic-co-glycolic acid (PLGA)/ intralesional/single dose 5 μg (0.2 mg/kg) of AmB in Day 30 (the model of established CL) | PBS | lesion size /lesion´s parasitic load determined by limiting dilution | 90 days after infection | At the end of follow up, lesions were 69% smaller, and the parasite burdens in the ear and draining lymph nodes were 97% and 87% smaller than control, respectively. |
| Alves, 2020 [22] | BALB/C | Tail base/ *L. (L.) major* | 8 | 40 days post-infection | AmB 1,5% with gallic acid (GA) 1.5%/ topical 50 μl twice daily for 21 days | AmB 3%; GA 3% and EA 3% each one topical with 50 μl twice daily for 21 days | lesion size/lesion´s parasitic load determined by limiting dilution | 14 days after the end of treatment | Amph B + GA and Amph B + EA had similar results to obtained with Amph B, GA, and EA/ Significant reduction in the parasitic load in animals treated with Amph B + GA e Amph B + EA. |
| | | | 8 | | AmB 1,5% with ellagic acid (EA) 1.5%/topical 50 μl twice daily for 21 days | | | | |
| Dar, 2020 [34] | BALB/C | Tail base/ *L. (L.) mexicana* | 5 | 28 days of infection | AmB-UDLs gel/ topical 8 mg/kg, twice daily for 28 days | Untreated group (Carbopol gel topical) and simple AmB gel | lesion size/lesion´s parasitic load determined by limiting dilution | 28 days after the treatment | Significant reduction of the lesion size, but not completely resolved/ substantial reduction in the parasite burden. |
| Fernández-Garciá, 2020 [23] | BALB/C | Tail base/ *L. (L.) amazonesis* | 4 | 35 days of infection | AmB transfersomes vesicles (TFs)/topical 20 mg once daily for 10 days | Untreated animals (control group); Glucantime (intralesionally 25 μL, 50 mg/kg) | lesion size/lesion´s parasitic load determined by limiting dilution | 56 days post-infection | The lesion size reduction was significant only 6 days after the end of the treatment/ Decrease in the parasitic load was similar to that observed with intralesionally administered Glucantime. |

*(Continued)*

**Table 1.** (Continued)

| Year, Author | Infection models | | | Treatment | | | Outcome | | |
|---|---|---|---|---|---|---|---|---|---|
| | Animal | Infection site/ *Leishmania* species of infection | Animals/ group | Start of treatment | Intervention/ administration route treatment schedule | Comparator | Methodology | Follow-up | Intervention X Comparator |
| Riaz, 2020 [35] | BALB/C | Tail base/ *L. (L.) major* | 4 | 14 days of infection | AmB nanostructured lipid carriers (NLCs)/ topical 50 μl for 10 days | Intravenous liposomal AmB (positive control) and no treatment (negative control) | lesion size/lesion´s parasitic load determined by limiting dilution | 24 days post-infection | No significant difference of the lesion size/ significant reduction in the parasitic load compared to the negative control. |
| Baharvandi, 2022 [30] | BALB/C | Tail base/ *L. (L.) major* | 10 | NR | AmB in microemulsion (ME) 0.4% topical, twice a day for 28 days (0.4 mg/day) | ME-Gel; AmB-Gel and Placebo topical/twice a day for 28 days | lesion size/spleen´s parasitic load determined by Real-Time PCR | 56 days post-infection | Remarkably smaller lesions/lower parasitic load compared to the placebo group. |

**NR**: not reported; **PCR**: polymerase chain reaction; **NPs**: nanoparticles; **PLGA**: poly (lactic-co-glycolic acid); **NLCs**: nanostructured lipid carriers

## Clinical studies

Only one randomized clinical trial (RCT) was identified [36]. Among the other clinical studies, only one presented a control standard treatment group [37], while two studies compared two different AmB regimes [38,39] and one was not comparative [40]. The main methodological characteristics of the studies are presented in Table 2.

Four out five studies were performed in the Old World, three in Iran [36,37,40] and one in India [38]. Only one study was carried out in the Americas (Colombia) [39]. The number of treated patients ranged from 22 to 93 individuals. Intralesional infiltration [38,40] and topical [36,37,39] administration was evaluated in two and three studies, respectively. Two studies evaluated formulations produced by pharmaceutical companies, such as Humax Pharmaceutical S. A/Colombia [39] and Razaak Arak Pharmaceutical Company/Iran [37]. The other studies evaluated new formulations produced in-house.

All studies included patients with active skin lesions and CL confirmation based on direct examination, culture, or polymerase chain reaction (PCR). Only one study defined cure as complete re-epithelialization of all lesions and complete disappearance of the induration [39]. In other studies, the definition of cure was variable, in general assumed in face of 75 to 90% of epithelialization/ involution of the lesion [36,38,40]. In one study, the definition adopted for cure was not stated [37].

Only one study reported the relapse definition adopted: 100% re-epithelialization by Day 90 and subsequent emergence of lesions by Day 180 [39]. The time for cure assessment varied widely among studies, ranging from 42 [37] to 240 days [36]. The follow-up length was in general also quite variable, ranging from 1.5 [37] to 13 months [40].

As shown in Table 3, the studies included predominantly adult patients, with an average age varying from 21 to 51 years. Overall, there was a balance between male and female participants in all studies [36–38,40], except in López et al., 2018 [39], where only male soldiers of the Colombian Army were included. Most of the lesions were located on the head and neck [36,40], followed by the upper limbs in two studies [38,39]. One study did not report the lesions' location [37]. The duration of symptoms before treatment varied from 4 to 12 months.

**Table 2. Main characteristics of the human clinical studies.**

| Year, author | Study design | Country (cases) | Treatment arms (patients) | Treatment scheme | CL case definition | Cure definition | Relapse definition | Cure assessment (days) | Follow-up (month) |
|---|---|---|---|---|---|---|---|---|---|
| Layegh, 2011 [36] | Comparative, randomized | Iran (n = 110) | Topical L-AmB [formulated from AmB deoxycholate] (50) | 3–7 drops twice daily, for 8 weeks | Positive skin smear or biopsy of lesions within less than 6 months | 75% decrease in the induration size | NR | D56 | 6 m |
| | | | Intralesional meglumine antimoniate [Glucantime] (60) | Once a week, until a fully infiltrated lesion to a maximum dose of 2 mL for 8 weeks | | | | | |
| Goyonlo, 2014 [40] | Noncomparative, nonrandomized | Iran (n = 93) | Intralesional AmB 2 mg/ml solution (93) | 0.1 to 0.3 mL once a week, for up 13 weeks | Positive skin smear or lesion biopsy and history of antimony resistance or side effects | More than 90% reduction in inflammation and indurations | NR | D84 | 1–13 m |
| López, 2018 [39] | Noncomparative, nonrandomized | Colombia (n = 80) | Topical AmB 3% [Anfoleish; H*umax Pharmaceutical S. A and PECET]* (80) | Topical three times daily, for 4 weeks | Positive skin smear or culture or PCR | Complete re-epithelialization of all ulcers and complete disappearance of the induration | Lesion that achieved 100% re-epithelialization by Day 90 that subsequently reopened by Day 180. | D90 and D180 | 6 m |
| | | | | Topical twice daily, for 4 weeks | | | | | |
| Goswami, 2019 [38] | Noncomparative, nonrandomized | India (n = 50) | Intralesional AmB solution 2.5 mg/ml (25) | Once a week, for 8 weeks | Positive skin smear for *Leishmania donovani* bodies | More than 90% reduction in size, induration, and ulceration; skin smear negative | NR | D84 | 6 m |
| | | | Intralesional AmB solution 5 mg/ml (25) | Once a week, for 8 weeks | | | | | |
| Khamesipour, 2022 [37] | Comparative, nonrandomized | Iran (n = 52) | Topical nanoliposomes with AmB 0.4% [*Razaak Arak Pharmaceutical Company]* (22) | Twice daily, for 4 weeks | Positive direct smear, culture and PCR | NR | NR | D42 | 1,5 m |
| | | | intralesional meglumine antimoniate plus cryotherapy (30) | One injection per week (total of 7), plus biweekly cryotherapy (3 or 4 sessions) | | | | | |

**NR**: not reported; **L- AMB**: liposomal amphotericin B; **AmB**: amphotericin B; **PCR:** polymerase chain reaction

Two studies identified the species of *Leishmania* [37,39]. Among them, *L. (V.) panamensis* was the most reported species (66/80) (López, 2018) [39], followed by *L. (L.) major* [37] and *L. (L.) braziliensis* [39], identified in 52/52 [37] and 12/80 [39], respectively (Table 3).

For topical administration of AmB, cure rates ranged from 30% [39] to 81.8% [37] (Table 4), and the combined cure rate was 45.6% [CI: 27.5–64 .8%; I2 = 79.7; p = 0.002] (Fig 2A), with very low certain of the evidence (S4 Table). In an attempt to explain the high heterogeneity, studies were stratified by regions, confirming the possible influence of Leishmania

**Table 3. Characteristics of the population treated with amphotericin B in human clinical studies.**

| Year, Author | Treatment arm/scam | Age (M_d ± SD or Age variation) years | Sex (male: female) | Duration of symptoms (months before therapy ± SD) | CL lesion site: n/N | Leishmania species characterization: n/N |
|---|---|---|---|---|---|---|
| Layegh, 2011 [36] | Topical L-AmB/twice daily | 20.54 ± 18.72 | 23:27 | 1.06 ± 0.31 | Head and neck: 26/50 (52%); Hand: 18/50 (36%); Leg and trunk: 6/50 (12%) | NR |
| | Intralesional meglumine antimoniate/once a week | 25.30 ± 15.70 | 21:39 | 0.96 ± 0.43 | Head and neck: 22/60 (36.6%); Hand: 32/60 (53.3%); Leg and trunk: 6/60 (10%) | NR |
| Goyonlo, 2014 [40] | Intralesional AmB/once a week | 20.81 ± 15.26 | 44:49 | <6: 19/93 (20.4%); 6–12: 46/93 (49.5%); >12: 28/93 (30.1%) | Head and neck: 68/93 (73.1%); Upper limb: 37/93 (39.8); Lower limb and trunk: 17/93 (18.3%) | NR |
| López, 2018 [39] | Topical Anfoleish/3 times a day | 24 (21–29 median) | 39:1 | NR | Head and neck: 6/40; Thorax: 5/40; Upper limbs: 25/40; Lower limbs: 8/40 | L. (V.) braziliensis: 6/40 L. (V.) panamensis: 33/40 |
| | Topical Anfoleish/2 times a day | 24 (21–29 median) | 39:1 | NR | Head and neck: 6/40; Thorax: 2/40; Upper limbs: 30/40; Lower limbs: 6/40 | L. (V.) braziliensis: 6/40 L. (V.) panamensis: 33/40 |
| Goswami, 2019 [38] | Intralesional AmB/2.5 mg/ml once a week | 33.00 ± 19.17 | 10:15 | <6: 15/25 (60%) >6: 10/25 (40%) | Head and neck: 11/25; Upper limbs: 20/25; Lower limbs and trunk: 9/25. | NR |
| | Intralesional AmB/ 5.0 mg/ml once a week | 28.79 ± 17.08 | 16:9 | <6: 18/25 (72%) >6: 7/25 (28%) | Head and neck: 9/25; Upper limbs: 10/25; Lower limbs and trunk: 9/25. | NR |
| Khamesipour, 2022 [37] | Topical L-AmB/twice a day | 28–51 years | 9:13 | NR | NR | L. (L.) major 22/22 |
| | Intralesional meglumine antimoniate/once a week | 14–60 | NR | NR | NR | L. (L.) major 30/30 |

NR: not reported; **L- AMB**: liposomal amphotericin B; **AmB**: amphotericin B

The cure and relapse rates are shown in Table 4. Only Lopéz et al., 2018 [39] reported cure at D180, which would be assumed to be the definitive cure rate according to Oliaro et al., 2013 [15]. However, no difference was observed among the cure rates pooled at D90 and D180 [39]. The study of Goswami et al., 2019 [38], was the only investigation reporting the cure rate at D56 and D84. Here, we considered the cure rate reported at D84, as this timepoint was the closest to D90.

species as a factor involved in the outcome, in addition to other local factors impacting the cure rates, based on the high heterogeneity remaining in the evaluation of Old-World studies (63.8%, CI: 24.3–90.7%; I2 = 87.3; p = 0.005) (Fig 2B) [16]. No heterogeneity was observed for the combined cure rate gathering the studies performed in the New World (31.3%, CI: 22.1–42.2%) [39], which, however, may be reflecting the bias of both being conducted by the same researcher.

**Table 4. Cure and relapse rates according to the intent-to-treat approach (human clinical studies).**

| Year, Author | Treatment arm/scam | Initial cure rate (D90) n/N (%) | Relapse rate % (at 6 months) | Adverse effects (n/N) |
|---|---|---|---|---|
| Layegh, 2011 [36] | Topical L-AmB/twice daily | D56 = 22/50 (44) | 0/22 | Mild pruritus around the lesions (5/50) |
| | Intralesional meglumine antimoniate/once a week | D56 = 29/60 (48.3) | 0/22 | Erythema and edema at the injection site (7/60); Hypersensitivity (1/60) |
| Goyonlo, 2014 [40] | Intralesional AmB/2 mg/mL once a week | D84 = 57/93 (61.4) | 4/57 | Local pain during injection (93/93); Prolonged pain (>30 min) (17/93); Fibrosis at the injection site (12/93); Local allergic reaction (1/93); |
| López, 2018 [39] | Topical Anfoleish/3 times a day | D90 = 12/40 (30) | 0/40 | Burning sensation, itching and rash (5/40); Mild and transitory elevation of transaminases (2/40) or creatinine (1/40); |
| | Topical Anfoleish/twice daily | D90 = 13/40 (32.5) | 0/40 | Burning sensation, itching and rash (2/40) |
| Goswami, 2019 [38] | Intralesional AmB/2.5 mg/ml once a week | D84 = 22/25 (88) | 0/25 | Pain during injection after no more than 30 min (25/25) |
| | Intralesional AmB/5.0 mg/ml once a week | D84 = 16/25 (64) | 0/25 | |
| Khamesipour, 2022 [37] | Topical L-AmB/twice a day | D42 = 18/22 (81.8) | NR | Burning sensation (2/22) |
| | Intralesional meglumine antimoniate | D42 = 15/30 (50) | NR | NR |

**NR**: not reported; **L- AMB**: liposomal amphotericin B; **AmB**: amphotericin B

Adverse events

For studies addressing AmB administered by intralesional infiltration, the cure rate ranged from 61.3% [40] to 88% [38] (Table 4), and the pooled cure rate was 69.8% [CI: 52.3–82.9%; I2 = 63.9; p = 0.06) (Fig 3A), notably superior to that for topical treatment with AmB. Using GRADE a very low certain to the evidence was verify for the cure using intralesional AmB. The overall cure rate at D90 for all local interventions gathered (topical and intralesional AmB) was 56.9% [CI: 41.1–71.4%; I2 = 82.8; p = 0.00], as shown in Fig 3(B).

Two trials directly compared topical AmB treatment with meglumine antimoniate intralesional infiltration, allowing meta-analysis. Assessing 162 patients and considering cure at D90, no difference was observed between interventions (OR: 1.7; 0.34–9.15, $I^2$ = 79.1; p = 0.00) (Fig 4). Based on GRADE assessment, the certainty of evidence for AmB versus meglumine antimoniate comparison was set as very low (S4 Table).

Concerning adverse events, reports of burning or itching were consistently documented in all four evaluations of topical AmB administration, regardless of whether the drug was applied twice or thrice a day [36,37,39]. In turn, systemic adverse manifestations, such as a mild and transient elevation of transaminases or creatinine, were exclusively observed when topical AmB was administered three times a day [39]. Importantly, in all instances, these values returned to normal levels after treatment, during a follow-up period of up to 6 months [39]. In the other included studies, systemic adverse events were not monitored. Furthermore, the same study has reported additional adverse events unrelated to the treatment, including chickenpox, flu, muscle pain, and gastrointestinal symptoms [39].

In the context of intralesional AmB administration, reports of local pain during injections were complained by all treated patients, regardless of the therapeutic regimen and but not leading to the treatment interruption in any case [38,40]. Notably, in the study evaluating the lowest concentration of intralesional AmB (2mg/mL), other adverse events such as prolonged pain (for more than 30 minutes), fibrosis at the injection site, and local allergic reactions [40] were also reported, while no systemic allergic reaction was observed.

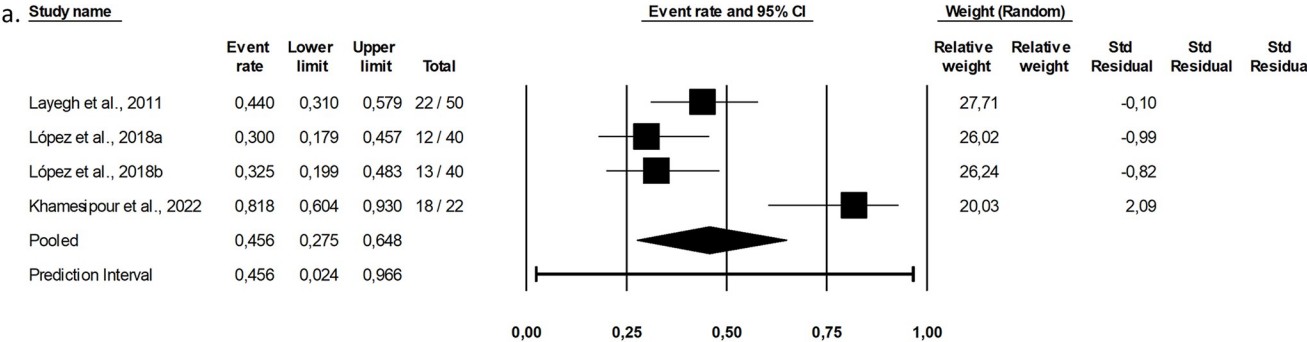

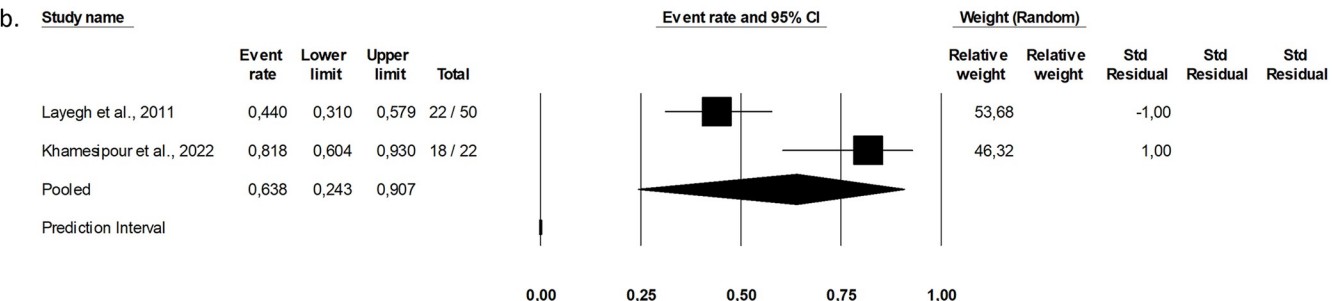

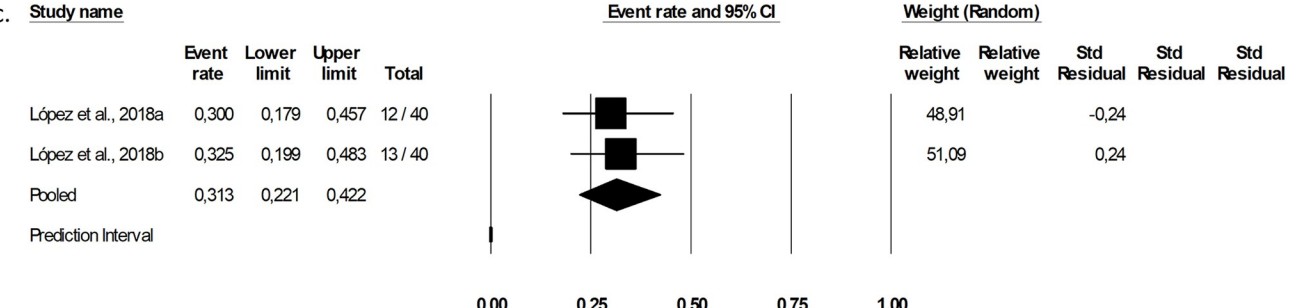

**Fig 2.** Pooled cure rate including all patients treated with topical amphotericin B (a), including only studies conducted in the Old World (b), including only studies conducted in the New World (c).

### Risk of bias in the included studies

The assessment of risk of bias is shown in Fig 5. Regarding preclinical studies, considerable homogeneity was observed. However, a high risk of bias was confirmed for selection, performance, and detection domains, given the absence of random allocation of animals and lack of blinding during cure assessment. A low risk of bias was observed in relation to the description of the result (Fig 5A).

Regarding nonrandomized clinical studies, it has been observed that the absence of a comparator is the factor that most impacts bias, affecting the domains of selection and comparability. Regarding the results, in general, a low risk of bias was verified (Fig 5B). The only one randomized clinical trial [39] was assessed using Rob 2.0 tool, with a high risk of bias being observed for most domains evaluated and some concerns regarding the randomization process and measurement of results (Fig 5C).

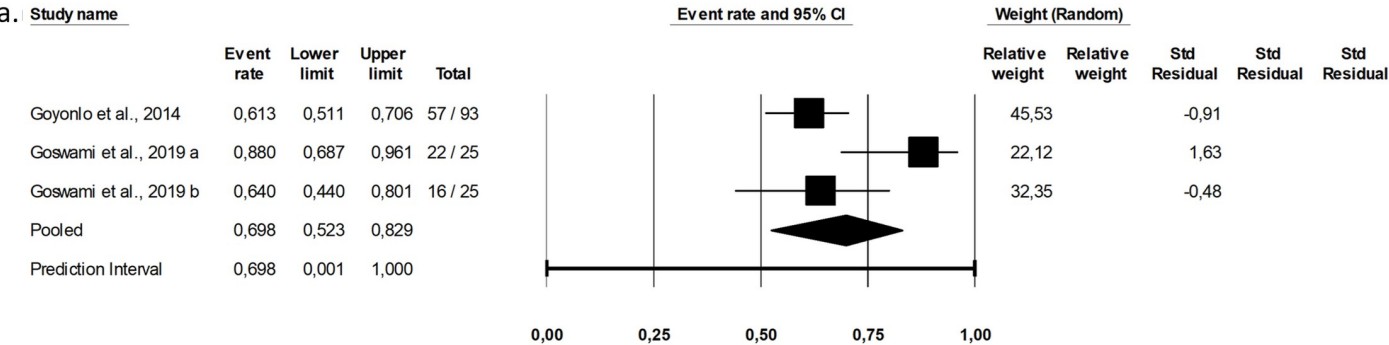

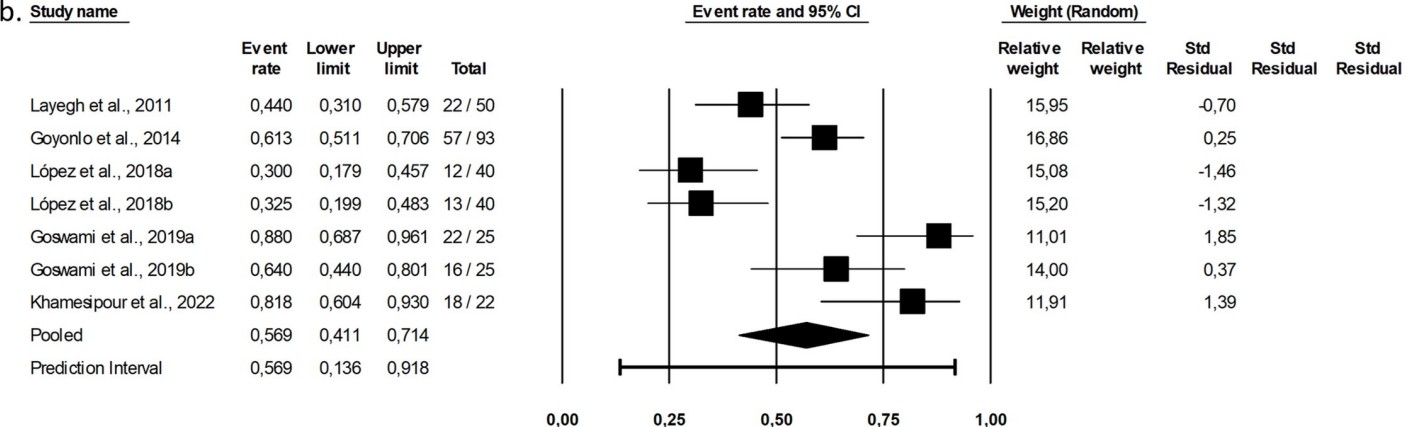

**Fig 3.** Pooled cure rate including all patients treated with intralesional (a) and topically (b) administered amphotericin B.

## Discussion

The first finding of this review was confirmation of the still limited experience with locally administered AmB interventions for CL: there are only 16 preclinical and five human studies published between 1984 and 2022. In addition to scarcity, the *Leishmania* species prevalent in Old World countries are more represented in the retrieved studies, and, in general, there is a

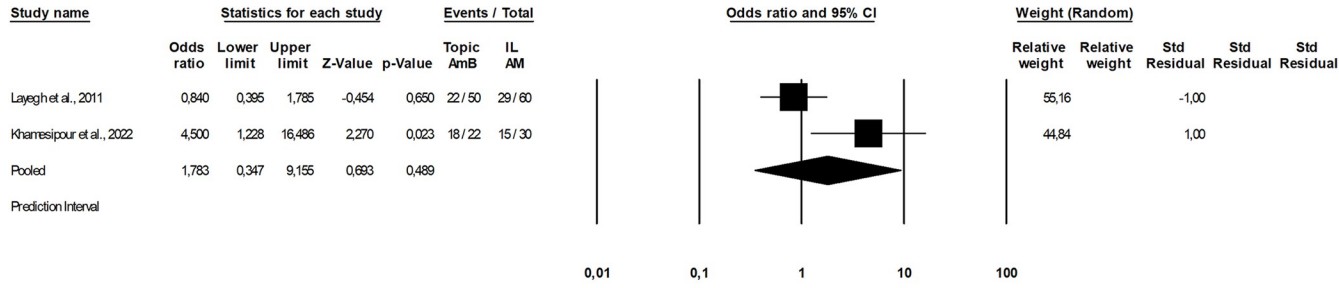

**Fig 4. Meta-analysis of studies directly comparing topical amphotericin B treatments and intralesional meglumine antimoniate.**

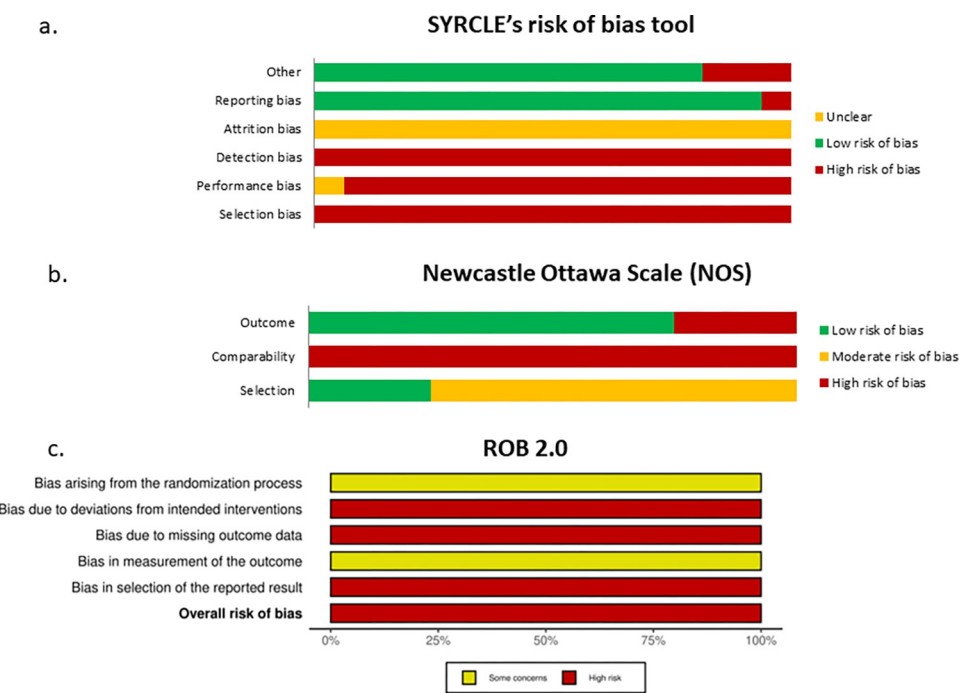

**Fig 5. Risk of bias observed in the included studies.**

lack of alignment between preclinical and clinical studies, that evaluated very different interventions, reflecting the absence of a strategic development plan for local treatment with AmB.

Treatment is still considered the main strategy for leishmaniasis control, given the lack of vaccines and difficulties in implementing actions focused on the vector. However, the therapeutic arsenal available is limited, either because of lack of investment in tropical infectious diseases or because of challenges related to the discovery and development of new effective drugs against this intracellular parasite. The local or systemic treatments currently recommended by the World Health Organization (WHO) and Pan American Health Organization (PAHO) are marked by many inconveniences, such as parenteral use, long-term treatments, and high toxicity, whether hepatic, cardiac, metabolic, or renal [9]. In this scenario, the repositioning and evaluation of alternative routes of administration of existing drugs emerges as a potentially useful strategy to expand the therapeutic options for leishmaniasis, aiming to overcome such limitations.

The switch in the antimony administration route in recent years represents a successful case of drug safety profile improvement in the CL field. With a cure rate like that observed with parenteral use [41], the intralesional infiltration approach is linked to greater schedule flexibility and a low risk of high-intensity or severe complications. The consolidation of the CL therapeutic modality based on antimony infiltration [9] possibly contributed to the increased interest in the local use of AmB. Based on this rationale, this review was proposed as the preparatory step for a future complete development plan for the local use of AmB, gathering existing preclinical and clinical trials. Preclinical trials represent an ideal starting point for the exploration of different formulations and schemes in different models. In preclinical studies, AmB (deoxycholate and liposomal) commercially available formulations are typically evaluated in association with carriers or controlled release systems, aiming to enhance drug dispersion and permeation. However, few studies assess whether these formulations and alternative routes can modify the bioavailability and, consequently, the toxicity of these drugs.

Deoxycholate AmB, when administered by intralesional route was detected in plasma with a concentration peak 12 hours post-infusion in animals. Nevertheless, when the drug was associated with poly(lactic-co-glycolic acid) (PLGA), AmB was not detected in the plasma during a 15 day post-infusion follow-up [21]. Significant reductions in lesion size in animals infected with *L. major* and *L. (L.) amazonenzis* were observed when using PLGA compared to AmB deoxycholate [21,33]. Sousa-Batista et al., 2019 [21] suggest that the low efficacy of deoxycholate AmB IL may be attributed to its rapid extravasation into the bloodstream. Very differently, in clinical studies, the pooled cure rate with AmB administered by intralesional route was estimated as 69.8% (52.3–82.9%) without a carrier, suggesting that even without exclusively local action this route of administration can be effective.

In preclinical studies, no hematological, renal, or hepatic toxicity was reported by using AmB associated with gallic and ellagic acid [22] and PMA [26] using topic administration. Subcutaneous concentrations of AmB capable of eliminating parasites were observed when this drug was encapsulated in ultradeformable lipid vesicles [23]. Furthermore, low concentrations of AmB were also detected in different organs after topical administration with ethanol, suggesting the possibility of using higher topical dose administration without systemic toxic effects [32]. The low systemic absorption, as well as the ability of AmB to penetrate the skin with the use of carriers, demonstrated in pre-clinical studies, are encouraging findings for carrying out human trials [29,32].

Amphotericin B, when associated with nanoparticles, has undergone assessment in four preclinical trials [27,29,33,35]. Among these, only poly (lactic-co-glycolic acid) nanoparticles loaded with amphotericin B deoxycholate, demonstrated a notable reduction in lesion size [33]. Regarding clinical trials, one study evaluated the topical treatment using nanoliposomes containing 0.4% amphotericin B and reported a cure rate of 81.8% (18 out of 22) for patients infected by *L. (L.) major*. However, this study involved a limited number of patients (n = 22) infected by *L. (L.) major*, a *Leishmania* species related to a relatively high rate of spontaneous cure.

A nice example of coordinated and sequential development was conducted by Layegh et al. (2011) [36] and Khamesipour (2022) [37]. In these cases, AmB liposomal topical formulations showed promising stability, diffusion, and efficacy results, as evidenced by preclinical investigations [29,32]. This success demonstrating the viability of a progressive and orderly construction of evidence that is not always observed [36,37].

Another aspect that deserves mention is, while cutaneous leishmaniasis (CL) is prevalent in 89 countries across the Eastern Mediterranean region, the Americas, and Africa [42], research efforts have predominantly focused on the Old-World disease, with a particular emphasis on species like *L. (L.) major* and L. *(L.) tropica*, as demonstrated in this review. As a summary of our findings, it is possible to state that there is limited representation of the Leishmania species and an absence of methodological standardization, coupled with a predominant qualitative approach among preclinical studies addressing local AmB treatments for CL, preventing are meta-analysis. Addressing this issue, a recently developed tool aims to standardize animal studies and enhance the reporting of experimental details [43]. Moreover, variations in formulations, treatment regimens, and notably, the initiation time of treatments—some even preceding the manifestation of lesions in certain protocols [25,26] constitute significant differences among studies.

Regarding clinical trials, most are not comparative, involving a small number of patients and presenting a high risk of bias in different domains. However, some useful observations maybe drawn from the experiences gathered thus far. It can be highlighted, among others, the observation of the lack of evidence for association between effect and number of applications of the ointment formulation addressed by López at al., (2018) [39], in the same way that a

higher concentration of the AmB solution infiltrated by Goswami et al., (2019) [38] did not affect the efficacy of the treatment.

Two topical interventions yielded therapeutic success in less than half of treated patients [36,39], while the intervention addressed by Khamesipour et al., (2022) [37] reached a cure rate of 81.8%. This discrepancy in results suggests that there may be a difference between formulations or, alternatively, it may have also been influenced by the cure definition adopted in the studies, not always clearly presented in the publication.

Many other factors also related to the biases possibly involved in this literature synthesis can be observed analyzing critically and in detail the studies. For example, the study with the highest cure rate enrolled a total of 22 participants infected with *L. (L.) major* (Khamesipour et al., 2022) [37] while the one with the lowest efficacy rate, conducted by López et al., (2018) [39] in Colombia, involved 80 patients infected with *L. (L.) panamensis* and *L. (V.) braziliensis*. In addition, this Colombian study adopted d the most demanding definition of cure based on a "complete re-epithelialization of the ulcers". Another noteworthy aspect is the differences between the spontaneous cure rates in the Old and New World, estimated at 60% for species such as *L. (L.) major* [44] and approximately 6% for *L. (V.) braziliensis* [45]. Although we cannot assume that the differences are due to Leishmania species, an information not always presented, it is possible to infer that different species may have contributed to the heterogeneity observed in these results.

The limitations inherent in the studies included in this analysis are noteworthy, a demanding caution in interpreting the summary cure measures we present. There are significant methodological differences, in addition to the very low certainty of the evidence. In the same way, even if only indirectly, a comparison between studies should not be made, but rather used as a strategy for identifying factors possibly associated with the direction of the observed effect. For example, Goswami et al., 2019 [38] and Goyolo et al., 2014 [40], despite having used similar schemes in terms of interval and dose (AmB 2.5 mg/ml once a week), presented different results, which may be related to the different *Leishmania* species present in Iran and India, respectively.

Based on the studies compiled in this review, it is not possible to identify one specific formulation or route for the local administration of AmB that is superior to others. Nevertheless, the information gathered is sufficient as a proof of concept on the feasibility of the local administration of AmB for Cl treatment. Among all the alternatives evaluated, the intralesional infiltration strategy based on the commercially available AmB appears to be the most viable option in the short term, considering all the regulatory requirements for the developing of a new drug. The use of new carriers that enable the dermal absorption of AmB also seems to be a promising strategy, which makes it even more necessary to include pharmacokinetic analyzes in the development plan of local approaches to leishmaniasis, in addition to efficacy and safety parameters.

## Supporting information

**S1 Table. PRIMA Checklist.**
(DOCX)

**S2 Table. Search strategies.**
(DOCX)

**S3 Table. Studies excluded during the full reading stage.**
(DOCX)

**S4 Table. Assessing the certainty of evidence using GRADE.**
(DOCX)

## Acknowledgments

We are grateful for support from the Programa de Pós Graduação em Ciências da Saúde of the Instituto René Rachou.

## Author Contributions

**Conceptualization:** Líndicy Leidicy Alves, Eliane de Morais-Teixeira, Gláucia Cota.

**Data curation:** Líndicy Leidicy Alves, Mariana Lourenço Freire.

**Formal analysis:** Líndicy Leidicy Alves, Mariana Lourenço Freire, Isadora Lana Troian, Gláucia Cota.

**Investigation:** Isadora Lana Troian.

**Methodology:** Líndicy Leidicy Alves, Mariana Lourenço Freire, Isadora Lana Troian, Eliane de Morais-Teixeira.

**Supervision:** Gláucia Cota.

**Validation:** Eliane de Morais-Teixeira, Gláucia Cota.

**Writing – original draft:** Líndicy Leidicy Alves, Mariana Lourenço Freire, Gláucia Cota.

**Writing – review & editing:** Líndicy Leidicy Alves, Isadora Lana Troian, Eliane de Morais-Teixeira, Gláucia Cota.

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
