## [Decision Letter · Decision Letter 0]

2 Oct 2023

Dear Dra. Alves,

Thank you very much for submitting your manuscript "Local amphotericin B therapy for cutaneous leishmaniasis: a systematic review" for consideration at PLOS Neglected Tropical Diseases. As with all papers reviewed by the journal, your manuscript was reviewed by members of the editorial board and by several independent reviewers. In light of the reviews (below this email), we would like to invite the resubmission of a significantly-revised version that takes into account the reviewers' comments. 

We cannot make any decision about publication until we have seen the revised manuscript and your response to the reviewers' comments. Your revised manuscript is also likely to be sent to reviewers for further evaluation.

Sincerely,

Ana Laura Grossi de Oliveira, Ph.D.

Guest Editor

Ricardo Fujiwara

Section Editor

Reviewer's Responses to Questions

**Key Review Criteria Required for Acceptance?**

**Methods**

-Are the objectives of the study clearly articulated with a clear testable hypothesis stated?

-Is the study design appropriate to address the stated objectives?

-Is the population clearly described and appropriate for the hypothesis being tested?

-Is the sample size sufficient to ensure adequate power to address the hypothesis being tested?

-Were correct statistical analysis used to support conclusions?

-Are there concerns about ethical or regulatory requirements being met?

Reviewer #1: Major revision: I have recommended the GRADE to evaluate the certainty of evidence.

Reviewer #2: The introduction of the study and its goal is clear. The methods are also clearly defined and in line with reporting guidelines for systematic reviews. There are, however, some concerns about the decision to conduct a meta-analysis and estimate pooled cure rates with very heterogeneous studies. Authors also described having to "arbitrarily assumed as the initial cure rate that assessed between D42 and D90."

Why did the authors decided for this approach instead of a qualitative synthesis of the available literature?

**Results**

-Does the analysis presented match the analysis plan?

-Are the results clearly and completely presented?

-Are the figures (Tables, Images) of sufficient quality for clarity?

Reviewer #1: Major revision: -The figures are not of sufficient quality for clarity, and need to be improved.

Reviewer #2: The results presented are clear and in line, again, with reporting guidelines. Authors have presented their findings in a very clear and succinct form. However, in line with the comment on the methods, the pooled estimates of cure rates with AmB treatment are difficult to interpret. The clinical studies included were conducted in populations infected with different Leishmania species, particularly OWCL and American species, which are known to differ in the rates of self healing and response to treatments. 

Regarding the presentation of adverse events, did any other of the studies (besides Lopez, 2018) assessed systemic effects (e.g., electrolytes, creatinine, transaminases, etc.)? If so, I would recommend stating in the manuscript whether systemic effects were assessed in these clinical studies and if none of them observed, which is different from lack of collection of systemic effects (e.g., no systematic measurement of blood chemistry and other markers).

In addition, it would be informative to describe the duration of these adverse events, particularly those related to intralesional administration. If not available, at least would be interesting to know if some of these are permanent. For example, table 4 describes prolonged pain (n=17/93) and fibrosis (n=12/93) at the injection site (Goyonlo,2014) in about 13-18% of treated cases, which would be important parameters to consider the feasibility of conducting further evaluations of intralesional AmB, given their potential impact in patient preference.

**Conclusions**

-Are the conclusions supported by the data presented?

-Are the limitations of analysis clearly described?

-Do the authors discuss how these data can be helpful to advance our understanding of the topic under study?

-Is public health relevance addressed?

Reviewer #1: The authors must be consider the GRADE analysis for conclusion.

Reviewer #2: Abstract: The conclusion stated in the abstract ('Amp.B is a promising strategy") is not well supported. Gicen that the abstract is a standalone section, I would suggest to present more details in the results of the abstract (e.g., safety or other clinical outcome) that supports the 'promising' nature of the strategy (or use other qualifications to describe the intervention).

Main text: The discussion section is clear and many of the limitations of the available literature are presented. Less is discussed about the decisions made by the author team and the implications on result interpretation, for example, the decision to ‘assume’ that evaluations were at D90 in the studies, and mostly, the decision of continuing with a meta-analysis given the high heterogeneity of the studies. 

In the pre-clinical studies, were there any PK or toxicity studies retrieved with the search strategy? All the included ones present outcomes related to cure, so, it would be good to know if the lack of PK/toxicity data was an artifact of the strategy selected to conduct the search.

In terms of conclusions, little could be inferred beyond the lack of standards and limited evidence for the use of AmB. This is a finding on its own and very relevant. However, it would be difficult to conclude, as authors have made for example, that continuing with intralesional administration of commercially available AmB is the best option. As mentioned in the results, more details on the safety would also help inform such a conclusion, but this is only briefly mentioned in the manuscript.

**Editorial and Data Presentation Modifications?**

Reviewer #1: (No Response)

Reviewer #2: The figures are barely legible. This may have been an effect during the creation of the pdf for review, but fig 2 and 2 are almost impossible to read (very small font size and blurred text).

**Summary and General Comments**

Reviewer #1: The comments were made in the attached file.

Reviewer #2: The study seeks to gather evidence in a very important topic, for which the number and quality of available published articles is limited. This is reflected in the results and discussion of the current systematic review. The main concern is the decision of authors to push for a meta-analysis in this context, which makes the results very difficult to interpret and potentially misleading. For example, the pooled cure rate for topical administration of AmB, was 45.6% [CI: 27.5–64.8%; I2 = 79.6), but the larger studies contributing to this number have very low cure rates (27%) while the smallest study, mostly in L. major (n=22), reported 81% cure. This is mentioned in the discussion, but one may argue that a qualitative synthesis of the available literature would allow to preserve these nuances when describing in the evidence. Given the objective of the study, a more detailed structure in the presentation of safety data (AEs), to see whether e.g., systemic AEs did not occur vs were not collected in the study, would be benefitial. 

Other comments to the manuscript are detailed in the respective sections.

PLOS authors have the option to publish the peer review history of their article (what does this mean?). If published, this will include your full peer review and any attached files.

Reviewer #1: Yes: IZABEL GALHARDO DEMARCHI

Reviewer #2: No
---

## [Decision Letter · Decision Letter 1]

20 Feb 2024

Dear Dra. Alves,

Thank you very much for submitting your manuscript "Local amphotericin B therapy for cutaneous leishmaniasis: a systematic review" for consideration at PLOS Neglected Tropical Diseases. As with all papers reviewed by the journal, your manuscript was reviewed by members of the editorial board and by several independent reviewers. The reviewers appreciated the attention to an important topic. Based on the reviews, we are likely to accept this manuscript for publication, providing that you modify the manuscript according to the review recommendations. 

Sincerely,

Ana Laura Grossi de Oliveira, Ph.D.

Guest Editor

Ricardo Fujiwara

Section Editor

Reviewer's Responses to Questions

**Key Review Criteria Required for Acceptance?**

**Methods**

-Are the objectives of the study clearly articulated with a clear testable hypothesis stated?

-Is the study design appropriate to address the stated objectives?

-Is the population clearly described and appropriate for the hypothesis being tested?

-Is the sample size sufficient to ensure adequate power to address the hypothesis being tested?

-Were correct statistical analysis used to support conclusions?

-Are there concerns about ethical or regulatory requirements being met?

Reviewer #2: The study objectives are clear and methodology adheres to the reporting standards for these studies. Authors have addressed most of the comments from the previous round.

Reviewer #3: The objectives of thi study articulated with a clear objective. The population clearly described and appropriate for the hypothesis being tested. The sample size is not large but sufficient to ensure adequate power to address the hypothesis being tested. Correct statistical analysis used to support conclusions. The paper is a review based on the results of clinical treatment, there are no concerns about ethical or regulatory requirements.

**Results**

-Does the analysis presented match the analysis plan?

-Are the results clearly and completely presented?

-Are the figures (Tables, Images) of sufficient quality for clarity?

Reviewer #2: The results are clear and in line with the analysis plan. 

Authors have now presented the findings by region and added comment on the GRADE low certainty of the evidence, which are welcome edits and provided important nuance to the interpretation of the results.

Regarding adverse events (table 4 and related texts), it is still not clear whether systemic adverse events were not inquired (i.e., no blood test or questions were performed as part of the study) in some of the studies instead of not reported. This distinction is very important to understand the systemic effects of the intralesional or topical administration of amphotericin B.

Reviewer #3: The analyses presented matched the analysis plan and the results clearly presented. The figures (Tables) are of sufficient quality.

**Conclusions**

-Are the conclusions supported by the data presented?

-Are the limitations of analysis clearly described?

-Do the authors discuss how these data can be helpful to advance our understanding of the topic under study?

-Is public health relevance addressed?

Reviewer #2: The discussion has improved and the limitations of the data analyzed are better presented. Some additional comments include:

Lines 340 - 341: please add the confidence intervals when describing the pooled estimate of the clinical studies.

Lines 349 - 351: please add references to this statement

Lines 372 - 373: Please verify the grammar of this sentence, it is not clear what is the meaning of: "It is and the comparison of studies"

Authors indicate that there is "lack of association" between the concentration and frequency of administration with the outcome. However, most of the studies are small and it is difficult to discern whether the lack of observed effect is is real or due to the small sample size. In this case one can argue that there is not evidence of an association (due to the size of the studies), rather than a lack of association.

Reviewer #3: The conclusions are supported by the data presented. The limitations of analysis clearly described.

**Editorial and Data Presentation Modifications?**

Reviewer #2: Figures are slightly better, however, figures 2-4 are still blurred in the pdf available for review and the font size is too small. Please revise the quality of the figures (not just that they meet a file format suggested by the journal, but that the font size and layout are legible).

Some of the tables have missing horizontal lines (e.g., Table 2 and 4)

Reviewer #3: (No Response)

**Summary and General Comments**

Reviewer #2: (No Response)

Reviewer #3: Leishmaniasis is still a disease epidemic in many countries, and effective therapeutic and preventive tools are still missing but urgently needed. In this manuscript the authors systematically reviewed the efficacy of treatment approaches with two AmB-based regimes on CL. The data were gathered mainly from studies carried in Iran, Colombia and India in the last 12 years. The cure rates significantly varied with very low certainty, which were likely determined by various factors including parasite species, subject physiological conditions and others. The review is comprehensive, but the significance of the study may be limited due to the lack of novelty of the treatment regimes.

PLOS authors have the option to publish the peer review history of their article (what does this mean?). If published, this will include your full peer review and any attached files.

Reviewer #2: No

Reviewer #3: No

Figure Files:

Data Requirements:

Reproducibility:

References

---

## [Editor Report · Decision Letter 2]

1 Apr 2024

Dear Dra. Alves,

We are pleased to inform you that your manuscript 'Local amphotericin B therapy for cutaneous leishmaniasis: a systematic review' has been provisionally accepted for publication in PLOS Neglected Tropical Diseases.

Best regards,

Ana Laura Grossi de Oliveira, Ph.D.

Guest Editor

Paul Brindley

Editor-in Chief

---

## [Editor Report · Acceptance letter]

9 Apr 2024

Dear Dra. Alves,

We are delighted to inform you that your manuscript, "Local amphotericin B therapy for cutaneous leishmaniasis: a systematic review," has been formally accepted for publication in PLOS Neglected Tropical Diseases.

Best regards,

Shaden Kamhawi

co-Editor-in-Chief

Paul Brindley

co-Editor-in-Chief
